# Recent Advances in the Reactor Design for Industrial Wastewater Treatment by Electro-Oxidation Process

**Jun Liu** [1,2], **Na Ren** [1], **Chao Qu** [1], **Shanfu Lu** [1], **Yan Xiang** [1] and **Dawei Liang** [1,*]

1 School of Space & Environment, Beijing Key Lab Bioinspired Energy Materials & Devices, Beihang University, Shahe Campus, Beijing 102206, China
2 China ENFI Engineering Corp., ENFI Equipment Branch, No.12 Fuxing Road, Haidian District, Beijing 100038, China
* Correspondence: liangdw@buaa.edu.cn

**Abstract:** Refractory organic wastewater mainly includes wastewater from papermaking, textile, printing and dyeing, petrochemical, coking, pharmaceutical and other industries, as well as landfill leachate and its membrane-treated concentrate. The traditional biochemical method is difficult to adapt to its harsh conditions such as high toxicity, high organic load and high salinity. Compared to other methods, the electro-oxidation (EO) process owns the attractive characteristics of being clean and eco-friendly, highly efficient and producing no secondary pollution. EO systems mainly include electrodes, a reactor, a power supply and other basic units. The design of reactors with different electrodes was the key link in the application of EO technology. This paper mainly reported the different configurations of electrochemical reactors (ECRs) for refractory organic wastewater treatment, and summarized the advantages and disadvantages of them, including reactor structure, flow mode, operation mode and electrode construction. Compared with traditional reactors, the improved reactors such as 3D-ECR achieve higher mass transfer efficiency by increasing the contact area between the electrode and the fluid. Additionally, it has a higher removal rate of organics and a lower energy consumption. Finally, the future perspectives of the treatment of refractory organic wastewater by ECRs is discussed. This paper is expected to provide a reliable scientific basis for the real application of EO technology in refractory organic wastewater treatment.

**Keywords:** wastewater treatment; refractory organics; electro-oxidation; reactor design

## 1. Introduction

Refractory organic wastewater is complex in composition and highly toxic, which has a significant impact on environmental and human health if it cannot be properly disposed of [1]. Therefore, high-efficiency technologies for the treatment of refractory organic wastewater are in urgent need. In many cases, it is difficult to achieve the ideal treatment effect by using the traditional biological method, and it needs to be performed in tandem with physical and chemical pre- and/or post-treatment [2,3]. Thus, the operation process is complex and the cost is relatively high. Over the past years, many advanced oxidation processes (AOPs) for refractory organic wastewater, including ozonation [4], photocatalytic oxidation [5], Fenton [6], electro-Fenton [7], photo-Fenton [8], persulfate-based oxidation [9] and electro-oxidation (EO) [10], have been deeply studied. The emerging AOPs can degrade or even mineralize the refractory organic compounds through hydroxyl radicals produced directly or indirectly [11]. Compared with other methods, the EO process owns the attractive characteristics of being clean and eco-friendly, highly efficient and without secondary pollution [12]. Therefore, EO has been a research hotpot in refractory organic wastewater treatment in the last several decades.

EO could be divided into direct oxidation and indirect oxidation according to the oxidation mechanism as illustrated in Figure 1. Direct oxidation of organic matter occurs

on the anode surface by the anodic generation of $\cdot OH$. Indirect oxidation is performed by using the anodic by-products as oxidants, such as $\cdot OH$, $O_3$ and $HClO$, to react with organic matter in aqueous solution [13].

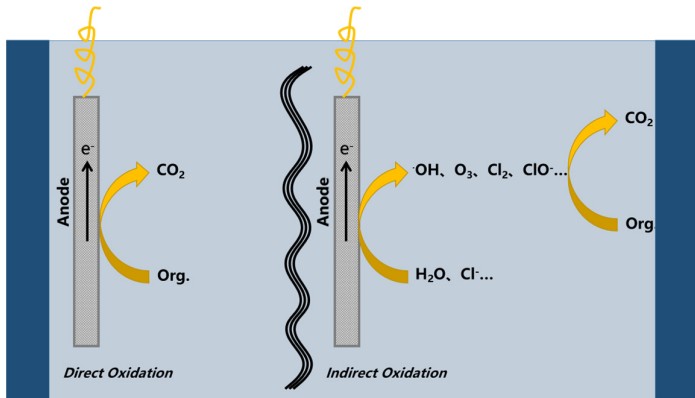

**Figure 1.** Schematic diagram of the reaction mechanism of an EO reactor.

The obstacle of EO for industrial application lies in the low current efficiency and high energy consumption [14]. Solving these problems could greatly advance EO technology. So far, the research directions of EO in wastewater treatment mainly focus on the advancement of anodic materials, which is targeting the improvement of the current efficiency for $\cdot OH$ generation and the stability or service life of electrode. Another focus is on the reactor configuration of EO. The optimization of reactor design could improve mass transfer coefficient, reaction kinetics and reduce energy consumption.

At present, the main anode materials include a metal electrode (e.g., Fe, Al, Pt) [15], graphite electrode [16], dimensional stable anode (DSA) [17] and boron-doped diamond (BDD) electrode in an EO system for wastewater treatment [18]. To overcome the deficiencies presented by the traditional graphite, platinum, lead-based alloy electrode and lead dioxide electrode, a titanium-based metal oxide electrode, also termed as DSA, is the most widely applied and has been developed for three decades [19]. However, there is a trade-off effect for the performance and service life of these electrodes. For example, $Ti/RuO_2$-$IrO_2$ is very stable and has long service life, but the oxidation efficiency is limited, while $Ti/SnO_2$-$Sb_2O_5$ and $Ti/PbO_2$ electrodes show great organic degradation performance but short service life limits their practical application [20]. A BDD electrode is a more robust electrode, because of its considerable chemical and electrochemical stability and the high oxygen evolution potential (OEP) to effectively produce $\cdot OH$ [21]. However, the price of the BDD electrode is too expensive for large-scale application.

In addition to the anode, the cathode can also remove the organic matter. The traditional cathodic hydrogen evolution reaction (HER) produces $H_2$, which can separate part of the colloidal organic pollutants in water through gas flotation [22]. The most widely used cathodes are stainless steel [23], nickel foam [24], etc. Recently, researchers also resort to carbon-based gas diffusion cathodes to produce $H_2O_2$ through the cathodic oxygen reduction reaction (ORR). $H_2O_2$ not only can directly degrade organic matter to some extent, but can also react with the anodic byproduct, such as $O_3$, $Cl_2$ or the added chemical $Fe^{2+}$, to produce $\cdot OH$ through the Fenton or Fenton-like oxidation mechanism [25]. Carbon-based materials with benefits of earth-abundant reserves, flexible structures, excellent electrochemical stability and high efficiency were considered as the optimal catalysts for $H_2O_2$ generation [26]. At present, carbon-based materials for a $H_2O_2$ generation cathode are graphite [27], carbon felt [28], carbon nanotubes (CNTs) [29], carbon/polytetrafluoroethylene (C/PTFE) [30] and a gas diffusion electrode (GDE) [31]. Moreover, GDEs are more popular because they could enhance the utilization efficiency of $O_2$, and even directly use air to produce $H_2O_2$ [32]. So far, over 300 review papers on electrode materials for EO

technology in wastewater treatment have been published. Thus, this study will not elaborate more on the electrode materials.

Reactor configuration is crucial of electrocatalytic degradation efficiency of pollutants and minimizing energy consumption for EO technology. The configuration design of electrochemical reactors (ECRs) has a direct impact on the fluid flow, mixing affect, heat transfer, mass transfer, etc. It has an especially significant impact on the mass transfer of reactant in solution, thus affecting the kinetic electrochemical reaction rate and electrocatalytic efficiency [33]. The main task of the reactor design is choosing the form and operation mode of ECRs. Then, it is necessary to calculate the fluid speed, temperature, pressure and the reactor volume according to the characteristics of the organic matter and electrochemical reactions. At the same time, economic benefits and environmental protection requirements should also be considered.

In recent years, the design of ECR configuration has improved greatly. The optimization direction is mainly focused on improving the mass transfer efficiency by increasing the contact area between the electrodes and the fluid. However, most of them still stay in the laboratory or pilot scale, which is not conducive to practical application.

Hence, this mini-review focuses on the configuration development and optimization of ECRs, and discusses advantages and disadvantages of different reactor design as well as the treatment efficiency of wastewater in the EO system. It may also provide theoretical support for the reactor design in fluid simulation, put forward the existing technology problems and the development orientation. It is hoped to provide a scientific basis for broadening EO in real applications for refractory organic wastewater treatment.

## 2. Electrochemical Reactors

For the design of the ECRs, several principles need to be considered. First, ECR design should meet the requirements of electrocatalysis. Second, the assembly of the ECRs should have high feasibility, indicating the reactor configuration, operation and maintenance should be as simple as possible; the electrode materials need to be easily obtained and inexpensive. Last, the reactor should have good versatility and be environmentally friendly [33]. The ECRs will be classified according to reactor structure, operation mode, flow mode and electrode configuration (Figure 2).

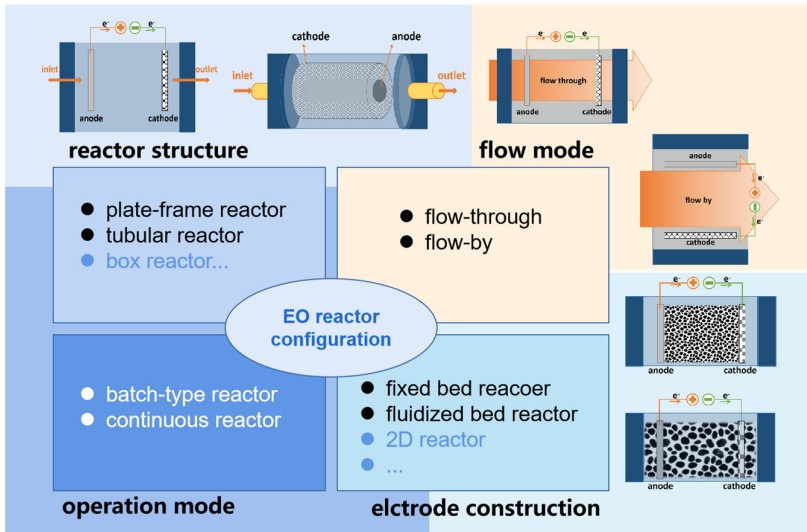

**Figure 2.** Reactor configurations of ECRs.

## 3. ECRs with Different Cell Architecture

The box reactor is the simplest structure of ECRs, applied with a plate electrode into a rectangular tank. It is widely applied in industrial electrolysis, organic synthesis, electroplating and other fields. A rounded beaker with nonuniform stirring pattern under the open system is usually utilized for wastewater treatment in lab, as this type of reactor shares the same principle as the box reactor [34]. However, it is difficult to apply a box reactor or rounded beaker on a large-scale, due to the low volume and time efficiency, the flow regime and mass transport effects that cannot be easily determined [35].

To improve the reactor mass transfer and spatiotemporal efficiency, reactors with special structures are developed, such as the plate-frame ECR (filter-press reactor) and tubular ECR. The plate-frame ECR is becoming popular due to its high performance in industrial applications, such as wastewater treatment, electrodialysis, energy storage process and electrosynthesis [36]. Compared with other reactors, the plate-frame ECR has the benefits of low pressure loss, a high space utilization rate and low cell voltage. The plate-frame reactor reduces energy consumption owing to the closer electrode space, so it shows great potential to realize EO in practical wastewater treatment. The tubular ECR uses tubular electrodes in the tank. Compared with the box reactor, it has more uniform current distribution and relatively large water polarity. Moreover, the efficiency of mass transfer and organic degradation efficiency could also be improved in tubular ECR. Therefore, the tubular reactor could further expand the research and application of EO technology.

### 3.1. Plate-Frame ECR

The plate-frame ECR, also called filter-press ECR as shown in Figure 3, is a commonly and extensively studied ECR [37]. The system usually consists of electrodes fitted in a parallel plate assembly and frames to hold the electrodes. The electrode formation varies, including non/porous or mesh plate, etc. The fluid flow may be continuous at a single or multiple inlets and outlets [38].

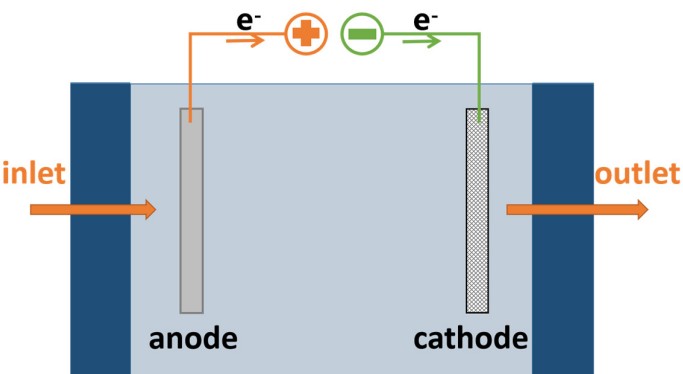

**Figure 3.** Schematic diagram of traditional plate-frame ECRs.

Except for the above mentioned merits, the plate-frame ECR also has relatively uniform potential and current distributions, which is good for reactor scaling up. The cell architecture is flexible, and is easy to combine with other technologies. The fluid flow and mass transfer are well-defined in a rectangular flow channel [39]. The plate-frame ECR has shown good performance in EO for wastewater treatment, and part of study results are shown in Table 1. Barbosa et al. (2018) studied an electrolytic flow process associated with the wastewater treatment using a filter-press reactor. The results showed that COD removal was 90% in 90 min [40]. Qu et al. (2019) studied a simultaneous EO and in situ electro-peroxone (ORR-EO) process for the degradation of leachate concentrate in a plate-frame ECR, obtaining 80% of TOC removal (from 250 mg/L to 50 mg/L) [41]. However,

the dead zone problem in plate-frame ECR is relatively serious, in which it cannot reach the ideal mixing state, and adversely affects the mass transfer of reactant in electrolyte.

At present, the researchers mainly improve it by rotating electrodes and application of turbulence promoters [33]. Quan et al. [42] designed a new plate-frame ECR for the treatment of landfill leachate. To keep the flow field uniform in the reactor, two perforated liquid distributors were mounted in the frame. The results showed that COD in the leachate could be reduced below 100 mg/L from 5000 mg/L after 60 min, reaching 98% of COD removal [42]. Nippatla and Philip [43] designed a rotating bipolar disc electrolytic plate-frame ECR for textile wastewater treatment. The turbulence was created, and mass transfer was enhanced, by electrode rotation. Thus, the COD removal was 4.8 times higher in a rotating electrode system compared to static electrodes [43]. Wachter et al. (2021) used a turbulence promoter in an electrochemical filter-press reactor to enhance the mass transfer, and the bisphenol S (BPS) degradation rate was increased approximately threefold to attain 98% of BPS removal [44].

**Table 1.** The treatment of different organic wastewater using plate-frame ECRs.

| Reactor Type | Anode | Cathode | Wastewater | Removal Rate | Ref. |
|---|---|---|---|---|---|
| filter-press reactor | $Sb_2O_5$-doped $Ti/RuO_2$-$ZrO_2$ | stainless steel | indigo carmine | 90% COD in 120 min | [45] |
| filter-press reactor | $Ti$–$Pt/\beta$-$PbO_2$ BDD | stainless steel | Estrone | 35% COD in 60 min 98% COD in 30 min | [46] |
| filter-press reactor | BDD | stainless steel | indanthrene blue dye | 91% color in 180 min | [47] |
| plate-frame reactor | $Ti/SnO_2$-$Sb_2O_5$ | GDE | landfill leachate | 80% TOC in 240 min | [41] |
| single-compartment filter-press reactor | BDD | stainless steel | paint wastewater | 90% COD in 90 min | [40] |
| plate-frame reactor with perforated liquid distributors | $Ti/RuO_2$-$IrO_2$ | stainless steel | solid waste leachate | 98% COD in 60 min | [42] |
| rotating bipolar disc plate-frame reactor | Fe | Fe | textile wastewater | 90% COD in 12 min | [43] |
| filter-press reactor with turbulence promoter | BDD | stainless-steel | BPS | 84% TOC in 265 min | [39] |

With the continuous improvement in the design of the ECR, the application of the plate-frame reactor is still limited by the low mass transfer efficiency and high energy consumption. Moreover, the reactor optimization process makes the structure of reactor more complex and increases the construction cost. Therefore, the balance between improving efficiency and elevating cost needs to be considered to scale up plate-frame ECR in EO applications.

### 3.2. Tubular ECR

The tubular reactor, also known as a plug-flow reactor, was developed in the oil smelting industry in the 1940s. It belongs to a continuous operating reactor. The traditional tubular ECR is composed of an anode rod nested inside the center of the outer cathode tube, called a concentric tubular reactor as shown in Figure 4.

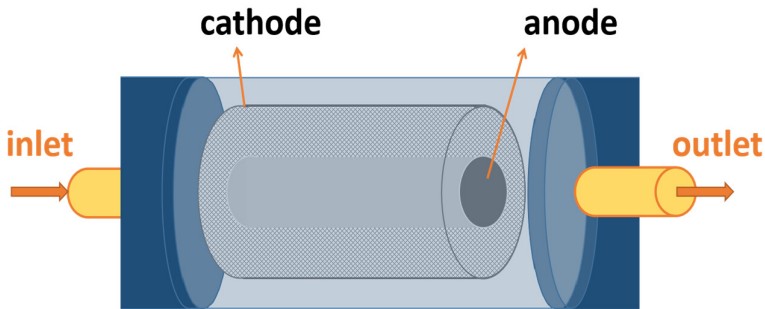

**Figure 4.** Schematic diagram of traditional tubular ECRs.

The tubular ECR has many advantages. Compared with a box reactor, tubular ECR electrodes have larger specific surface areas. It is easier to transfer the heat generated due to the ohmic loss in the electrolysis process to the external environment to avoid the accumulation of heat inside the reactor and provide long-term stable operation of the reactor. Tubular ECRs could reduce the reaction dead zone. The flow rate of fluid in the plug-flow reactor is faster, thus, it could decrease the volume of the reactor and improve the wastewater treatment efficiency. Additionally, the tubular reactor is convenient for combination and connection. The tubular ECR has been extensively and successfully applied in organic wastewater treatment, especially for high strength organic wastewater, such as dyeing, papermaking, food manufacturing, pharmaceutical wastewater, etc. Part of the research progress was listed in Table 2.

Körbahti et al. [48] investigated the treatment of dyeing wastewater in a tubular ECR, and obtained the respective COD, color and turbidity removals of 44.3%, 86.2% and 87.1% in 6 h. Moreover, the reactor maintains a high mass transfer coefficient of $3.62 \times 10^{-6}$ m/s [48]. Ibrahim et al. (2014) used a tubular ECR with a $Ti/RuO_2$ anode for the degradation of petroleum refinery wastewater, and complete removal of total petroleum hydrocarbon was achieved in 24 min with 85% of COD removal; they also obtained good reuse of petroleum refinery wastewater [49].

Although the performance of tubular reactors was satisfactory, it still has drawbacks. For example, the current density was not evenly distributed in a tubular reactor as the surface area of cylindrical anode and cathode are not equal, which needs optimization. The optimization of the reactor is mainly conducted from two aspects. One is to increase the contact area of the electrode with the fluid by changing the configuration of the electrode, such as mesh plate electrodes, porous electrodes and membrane electrodes. Another one is to reduce the internal dead zone of the reactor by changing the flow rate or pattern of fluid, according to the simulation results of the flow regime by computational fluid dynamics (CFD).

To deal with the problems of uneven current distribution, Li et al. [50] developed a tubular ECR with mesh plate electrodes and the solution flowed horizontally through the meshed plate electrodes. Moreover, CFD and particle image velocity (PIV) technology were used to simulate the internal flow characteristics. The results showed the mesh plate electrode could significantly promote the flow homogenization and reduce the back-reflux area in the reaction region by 50% [50]. The conclusion was coincident with Ibrahim's study results [51]. Wang et al. [52] presented a tubular ECR based on a vertical-flow reactor with mesh plate electrodes. The results showed that the turbulent intensity increased by 200% around the electrode surface. The mass transfer coefficient of the novel tubular reactor was more than twice of that of the traditional tubular reactor [52]. The same type of reactor was reported on in other research, which demonstrated that the increase of the anode effective area was one of the main factors to enhance degradation efficiency of wastewater [53].

For improving the turbulent intensity of reactors, Guo and You [54] presented a tubular ECR based on spiral flow caused by a turbulence promoter for degradation of methylene blue in wastewater. The results showed that the mean flow rate and turbulent intensity of the new tubular reactor were increased by 500–700% and nearly 200%, respectively, compared to the traditional one [54].

**Table 2.** Treatment of different organic wastewater by tubular ECR.

| Reactor Type | Anode | Cathode | Wastewater | Removal Rate | Ref. |
|---|---|---|---|---|---|
| traditional tubular reactor | Pt/Ti | carbon fibre | dyeing wastewater | 44.3% COD in 6 h | [48] |
| traditional tubular reactor | $Ti/RuO_2$ | stainless steel | petroleum refinery wastewater | 85% COD in 24 min | [49] |
| tubular reactor in batch with recirculation mode | $Ti/Ti_{0.7}Ru_{0.3}O_2$ | stainless steel | dye waste water | 75% of Acid red 87 | [55] |
| tubular reactor with mesh plate electrodes | $TiO_2/RuO_2$ | Ti | Evans blue dye | 80% COD in 100 min | [56] |
| tubular reactor with porous electrodes | $Ti/SnO_2$-Sb | stainless steel | pyridine wastewater | 86% TOC in 180 min | [57] |
| dual tubular membranes reactor | $Ti/IrO_2$–$Ta_2O_5$ | CB-PTFE modified graphite | tricyclazole | 79% COD in 20 min | [58] |
| tubular reactor based on spiral flow | $Ti/Ti_4O_7$ | stainless steel | methylene blue simulation wastewater | 50% TOC in 105 min | [54] |

At present, the research of tubular ECRs for refractory organic wastewater treatment remains in laboratory. The influencing factors on electrochemical reactions, and the process mechanism have not been fully understood, which needs further study to provide a more reliable theoretical basis for practical applications. At the same time, it is necessary to carry out large-scale reactor design. Researchers could analyze the flow conditions and the influence of chemical reactions on the flow by CFD and optimize the operating parameters accordingly.

## 4. Operation Mode of ECR

ECR can be operated in batch and continuous modes. The wastewater treatment in batch ECR is generally operated in a cycling pass in a fixed period of time, or until the organics are degraded enough to meet the discharging requirement. Batch treatment is normally suitable for small-scale or intermittent supplies of wastewater, and has the merit of a high impetus for the degradation kinetic rate, as the initial concentration of wastewater is high. However, batch-type reactors have the problem of limited treatment capacity. A continuous ECR is generally referred to as a "plug flow" reactor that operates in a single pass mode, with the reactant (organics) in influent contact with the electrodes in parallel or in series. From the engineering point of view, continuous mode is more preferred, as continuous ECRs provide major advantages such as faster reactions, cleaner products and safer and easier to scaling up, which outcompetes the limitations of batch reactors for large-scale wastewater treatment.

## 5. The Flow Mode and Electrode Configuration of ECRs

The ECR can be operated in flow-through and flow-by mode, according to the flow mode and the electrode configuration, as shown in Figure 5.

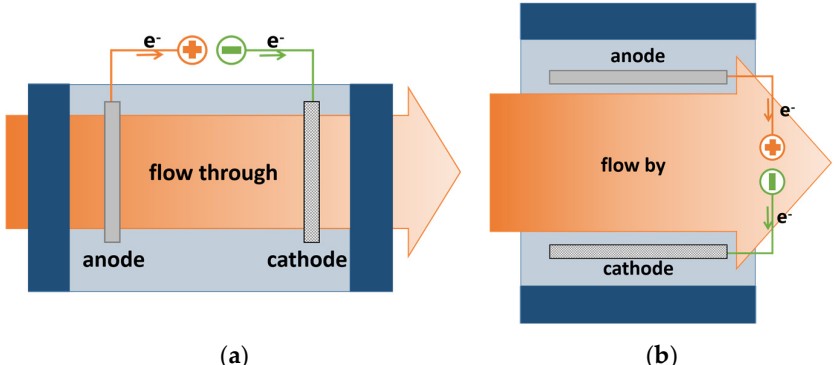

(**a**)　　　　　　　　　　　　　(**b**)

**Figure 5.** Schematic diagram of flow-through (**a**) and flow-by (**b**) mode of ECR.

### 5.1. Flow-By Mode

The flow-by mode of the ECR refers to when the flow direction of fluid is parallel to the electrodes, so that the aqueous solution cannot cross the electrode. Therefore, the electrodes morphology selected are mostly plate electrodes. As known, chemical and mass transfer processes could limit contaminant degradation rates. Due to the high reactivity of ·OH, the reaction with pollutants occurs in a narrow zone (<1 μm) on the anode surface, but the diffusion boundary layer of the anode surface is over 100 μm in the flow-by mode [59]. Thus, the limited mass transfer hindered by the boundary layer results in low treatment efficiency and high energy consumption.

Changing the fluid flow rate could improve the mass transfer. The flow rate of the liquor in the reactor determines whether the flow pattern is laminar or turbulent. This distinction determines the residence time of reactants in the reactor and on the surface of the electrodes. Rafael Machado Reis et al. (2013) reported on the degradation of dipyrone in a flow-by ECR with a laminar (50 L h$^{-1}$) or turbulent (300 L h$^{-1}$) regime, and a 92.5% TOC removal rate was achieved at an electrolyte flow rate of 300 L h$^{-1}$ [60]. However, recently, more research has resorted to high active surface area electrodes, which are flow-through mode electrodes, to improve mass transfer.

### 5.2. Flow-Through Mode

Flow-through mode refers to when the fluid direction is perpendicular to the electrode. Thus, flow-through ECR requires porous or mesh electrodes and the solution passes through pores 0.1–1.0 mm wide [61]. The flow-through mode could enhance the mass transfer of reactant by suppressing the boundary layer, thereby improving the degradation efficiency of different pollutants [59]. It is testified that the reaction rate constants of flow-through and flow-by modes were 0.0238 min$^{-1}$ and 0.0115 min$^{-1}$ according to the experimental data. The result means that flow-through mode ECR could drastically reduce the thickness of the diffusion boundary layer [62]. Moreover, the porous structure could improve the current efficiency by providing more reaction sites and reducing applied current. Therefore, it is beneficial to improve the utilization efficiency of electrons and increase the electrode lifetime [63].

Recently, the reported flow-through electrodes include porous or mesh anodes made of CNTs, Ti4O7, Ti/PbO2, Ti/RuO2 and Ti/SnO2-Sb. The pore size of the electrode is one of the key parameters for electrochemical reactions in an EO system. Chen et al. (2020) stud-

ied the pseudo-first-order rate constants (k) of MP-Ti ENTA/SnO$_2$-Sb$_2$O$_3$, which were estimated to be 0.38, 1.63 and 1.24 min$^{-1}$ with the nominal pore sizes of 10, 20 and 50 μm, respectively, in a flow-through ECR system [64].

To obtain an efficient and low-energy ECR for wastewater treatment, Pérez et al. [65] developed a microfluid flow-through ECR with a narrow internal electrode space and flow-through electrodes. Compared with a commercial flow-by ECR, the microfluid flow-through ECR required between 4- and 10-fold less current and from 6 to 15 times less energy consumption [65]. Yang et al. [63] constructed an electro-peroxone membrane filtration (EPMF) system based on ·OH production, and enhanced mass transfer with a CNT membrane cathode operated in flow-through mode. The results showed that the kinetic rate constant (*k*) was 1735.30 × 10$^{-3}$ s$^{-1}$ for ibuprofen (IBU) removal. In contrast, the value of *k* was only 16.4 × 10$^{-3}$ s$^{-1}$ in traditional batch reactors [63]. Zeng et al. [66] studied the efficient electro-oxidation of sulfamethoxazole (SMX) by a TiO$_2$ nanotube arrays-based electrocatalytic flow-through membrane, and the SMX removal efficiency of the flow-through mode was increased by 71%. It is verified, the superiority of a porous flow-through anode [66]. Zhang et al. [67] developed a flow-through ECR with a Ti$_4$O$_7$-based membrane electrode. It proved that this flow-through mode could enhance the mass transfer with a high tetracycline degradation efficiency (97.24%) and low energy consumption (0.18 kWh/g DOC) [67]. Maldonado et al. (2021) developed a boron-doped diamond (BDD) anode flow-through ECR for the electro-oxidation of perfluoroalkyl acids (PFAAs) in landfill leachates, and obtained a 73.5% of the total PFAAs removal after a 2 h reaction [68]. Ormeno-Cano and Radjenovic (2022) studied electrochemical degradation of antibiotics using flow-through graphene sponge electrodes. The results showed the removal of antibiotics was above 80% observed [69]. Chen et al. [62] developed a high-efficiency EO system in flow-through mode based on macroporous enhanced TiO$_2$ nanotube array/SnO$_2$-Sb (MP-Ti-ENTA/SnO$_2$-Sb) anode for treatment of reverse osmosis concentrates. The COD removal efficiency of the flow-through ECR was higher than 70% after 60 min, while the flow-by ECR required 120 min and used more energy [62].

In summary, compared with flow-by ECRs, flow-through ECRs show better performance in the electro-oxidation of organics and exhibit broader application prospects for wastewater treatment.

## 6. ECR Classification according to the Electrode Configuration

The ECR electrode configuration is divided into two-dimensional electrode (2D) and three-dimensional electrode (3D) configuration, according to the presence or the absence of particle electrodes in the reactor. A 3D-ECR consists of granular electrode materials packed in the middle of the traditional 2D flat plate electrode, and operated under a flow-by or flow-through mode. The particle electrodes can be polarized by an external electric field to form a large amount of bipolar micro electrodes [70]. The electrochemical reaction can occur on the surface of each particle. Thus, these particle electrodes extend surface area and enhance the mass transfer of the reactants in 3D-ECR [71].

The mechanism of organic degradation in 3D-ECR depends on the type of organic matter, electrode materials and particles as shown in Figure 6. The anodic and cathodic reactions are consistent with 2D-ECRs and the main redox reaction still occurs on the surface of the electrodes. The particle electrodes, including granular activated carbon (GAC), metal particles, carbon aerogel (CA) and modified kaolin, have good electrical conductivity and large surface areas. In the reaction, the particle electrodes are polarized to generate ·OH to degrade pollutants. Moreover, the particle electrodes would also remove some pollutants through adsorption [72]. It could improve the degradation efficiency by improving the concentration of pollutants at the particle electrode interface [73]. It is proven that the degradation rate of pollutants is mainly related to the adsorption performance of the particle electrode in a low voltage system [74]. Therefore, the degradation rate of organic matter in the whole reactor is greatly improved.

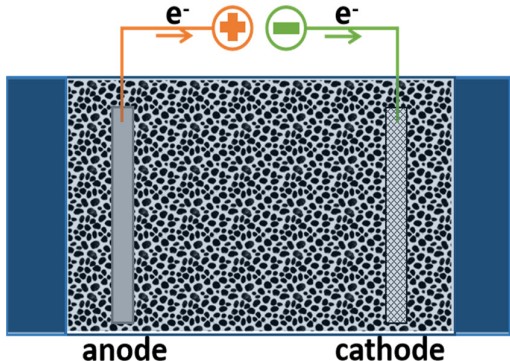

**Figure 6.** The structure of 3D-ECR.

The configurations of 3D-ECR are varied according to whether the particles are filled in fixed or fluidized mode. The fixed bed ECR and fluidized bed ECR were as shown in Figure 7.

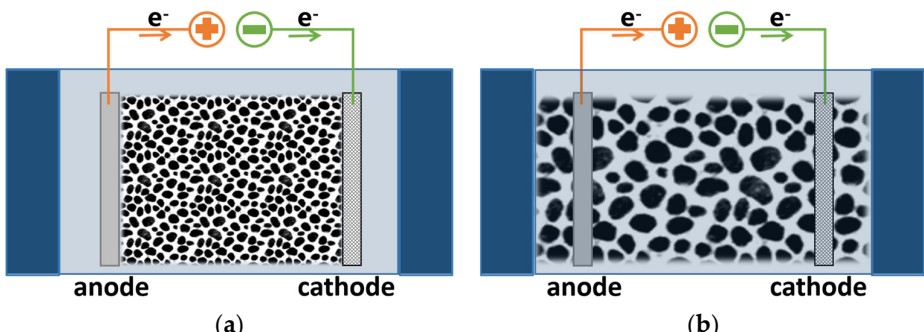

**Figure 7.** 3D fixed bed mode ECR (**a**) and 3D fluidized bed mode ECR (**b**).

### 6.1. Fixed Bed 3D-ECR

The fixed bed electrode has high spatiotemporal efficiency, area–volume ratio and current efficiency as well as easy controlling. The particles electrode in a fixed bed 3D-ECR hardly gain evenly distributed current and potential. To promote the mass transfer, magnetic stirring or air flow stirring is required. However, it is easy to form a short-circuit current and cake the particle electrode due to fixed bed mode, and thus, directly affect the current and energy efficiency of ECRs. An approach for reducing the short circuit problem in fixed bed ECRs is using particle electrodes with different densities, which are distributed evenly in the reactor [75].

### 6.2. Fluidized Bed 3D-ECR

The fluidized bed 3D-ECR provides a larger electrode activation area and high mass transfer rate, because the strong disturbance of the particles effectively reduces the concentration polarization of the electrolyte on the particle surface. However, uneven distribution of the current, and potential generated because the particle electrodes are not in close contact with one another can lead to reduced treatment efficiency. Kitaev and Rubanovich [76] developed that a reactor equipped with a mixer, consisting of an electric rotating shaft with a plurality of turning vanes arranged along the longitudinal axis, which has the advantages of saving energy. The power consumption per cubic meter of wastewater was approximately 10~12 kWh with organic treatment efficiency of 95–98% [76].

The 3D-ECR system has been widely used in the treatment of various wastewater, such as phenolic compounds, coking, oils, etc., as summarized in Table 3. Ni et al. [77] compared the TOC and COD degradation in Rhodamine B wastewater using a cycling-flow 2D-ECR and a 3D-ECR system. The results showed that the removal efficiencies of TOC and COD in a 3D-ECR were both approximately twice as those in 2D-ECR [77]. Wei et al. [78] investigated a 3D-ECR with GAC and a porous ceramsite particle as the electrode for pretreatment of heavy oil refinery wastewater. Results showed that the removal efficiencies of COD, TOC and toxicity units were 45.5%, 43.3% and 67.2%, respectively, and the ratio of BOD/COD was increased from 0.10 to 0.29, which is beneficial for further biological treatment [78].

The conventional particle electrodes, such as GAC, showed a modest degradation ability for pollutants and relatively high chemical stability during the EO process for wastewater treatment. However, these particle electrodes do not have enough catalytic activity to completely degrade the contaminants adsorbed on their surface, and the pores of the particle electrodes are easily clogged by contaminants during the reaction, so it will influence long term operation performance of wastewater treatment [79].

In recent years, it became a hot spot to enhance the EO performance of 3D-ECRs focusing on the surface modification of a particle electrode by loading catalyst. Li et al. [80] prepared GAC doped with $TiO_2$-$SiO_2$ oxide ($TiO_2$-$SiO_2$/GAC) as particle electrodes in a 3D-ECR for dyeing wastewater treatment. The results showed the $TiO_2$-$SiO_2$/GAC particles had a lower specific surface area and pore volume than the original GAC; this means that $TiO_2$-$SiO_2$ covered part of the outer surface and the internal pores of GAC. It could provide more probability in the degradation of the pollutants adsorbed into the pores [80]. Chen et al. [81] studied a 3D-ECR using the Sn/Sb-Mn-GAC particle electrodes to effectively treat 4-chlorophenol in wastewater. The oxygen evolution potential (OEP) of the Sn/Sb-Mn-GAC particle electrode was much higher than that of GAC, which can produce more oxidizing radicals and a higher removal rate; a rate of 96% in 60 min was obtained [81].

**Table 3.** Treatment of different organic wastewater by 3D-ECR.

| Reactor Type | Electrode Materials | Wastewater | Treatment Results | Ref. |
|---|---|---|---|---|
| 3D fluidized bed ECR in batch | Anode: stainless steel Cathode: stainless steel Particle electrode:GCA | Phenol | COD removal of 60% at an airflow of 5 min$^{-1}$ and voltage of 30 V | [82] |
| 3D fixed bed ECR | Anode: graphite Cathode: stainless steel Particle electrode: CAs | dye wastewater | decolorization ratio was 98% (pH = 2.8, voltage = 20 V, treatment time = 30 min, air flow rate = 0.4 L min$^{-1}$) | [83] |
| recycling-flow 3D-ECR | Anode: Ti/RuO$_2$/TiO$_2$ Cathode: Ti/RuO$_2$/TiO$_2$ Particle electrode: Columnar GAC | Rhodamine B wastewater | COD removal of 86.9% at HRT = 20 min, initial concentration = 100 min, volume = 500 mL, pH = 7, Na$_5$O$_4$ = 2 g/L and voltage = 5 V. | [77] |
| pilot-scale plate-frame fluidized bed ER | Anode: Ti/RuO$_2$-IrO$_2$ Cathode: Ti/RuO$_2$-IrO$_2$ Particle electrode: GAC | coking wastewater | removal 94.4% of COD and 76.2% of TN at a low EC of 0.22 kWh/kg COD and 4.69 kWh/kg TN. | [84] |
| 3D multi-phase ECR | Anode: graphite Cathode: graphite Particle electrode: Fe | petroleum refinery wastewater | COD removal of 92.8% at pH = 6.5, cell voltage = 12 V and fine Fe particle = 4 g and air flow rate = 1.5 L/min | [85] |
| 3D-ECR | Anode: DSA Cathode: Ti Particle electrode: Sn/Sb-Mn-GAC | 4-chlorophenol | 4-chlorophenol removal of 96.13% in 60 min at Na$_2$SO$_4$ concentration of 2 g·L$^{-1}$, electrode distance of 2 cm, current intensity of 2 A, and particle dosage of 14 g. | [81] |
| 3D-ECR | Anode: Ti/RuO$_2$-IrO$_2$ Cathode: stainless steel Particle electrode: TiO$_2$-SiO$_2$/GAC | dye wastewater | decolorization rate and COD removal efficiency are 83.20% and 48.95% at electric field intensity = 1 V/cm, TiO$_2$-SiO$_2$/GAC = 200 g/L, Na$_2$SO$_4$ = 0.07 mol/L, pH = 3 in 120 min. | [80] |
| 3D-ECR | Anode: Pt Cathode: Pt Particle electrode:Pd-Fe/Ni | Dimetridazole | The dimetridazole removal of 96.5 at current density = 31 mA/cm$^2$, C$_0$ = 50 mg/L, pH = 3, air flow rate = 1.0 L/min | [86] |
| 3D batch type ECR | Anode: Ti/Sb$_{0.1}$Sn$_{0.9}$O$_2$ Cathode: stainless steel Particle electrode: GAC and ceramsite particle (PCP) | oil refinery wastewater | removal of COD, TOC and toxicity units were 45.5%, 43.3% and 67.2% (GAC percentage = 75%, current density = 30 mA/cm$^2$, pH not adjusted and treatment time = 100 min) | [78] |

So far, refractory organic wastewater treatment with EO has achieved good results in the laboratory, but the cases of large-scale EO treatment are still lacking. The reasons are as follows: (1) The kinetic and thermodynamic processes of adsorption, desorption and electrochemical oxidation that occurred on the 3D electrode surface are poorly defined. (2) The electrocatalysis mechanism of organics in 3D-ECR is more complex than in 2D-ECR, and the wastewater treatment efficiency is influenced by many parameters, such as the cell structure, bed expansion rate, voltage, current density, reaction time, electrode material and reactant concentration, etc.

Increasing the voltage or current density to a certain extent is beneficial to pollutant degradation, but it also increases energy consumption. It is necessary to balance the processing effect and energy consumption [87]. Furthermore, increasing the amount of parti-

cle electrodes would result in more particle electrodes, but highly packed particle electrodes would occupy much space in the 3D-ECR, leading to reduce the amount of wastewater treatment [88].

The 3D-ECR system deserves comprehensive development due to its unique performance. In addition to the research on electrode materials for refractory organic wastewater treatment [89], the main research trend in the direction of ECR lies in optimizing and expanding the design of 3D-ECR. During the continuous operating process, the system may lose the ability to degrade the pollutants due to the accumulation of the pollutants in the reaction system. Optimizing the operating parameters and maintaining the stable operation of the 3D-ECR system has great commercial value, which is greatly significant for future development.

## 7. Conclusions and Future Perspectives

The design and optimization of ECR configuration plays a crucial role in the development of EO technology. The designs of ECRs are diverse and depend on different objectives, as well as the shape and characteristics of the electrodes in wastewater treatment. In this review, the key points were discussed as follow:

(1) Compared with the plate-frame ECR, the tubular ECR has less dead zone and a higher mass transfer efficiency due to the larger specific surface areas of electrodes. The mass transfer efficiency is the one of most important factors for degradation efficiency of wastewater in ECRs. The optimization reactors enhance degradation efficiency of wastewater mainly by changing the shape of electrodes or increasing fluid flow rate.

(2) The limited mass transfer hindered by boundary layer in flow-by ECRs. Additionally, flow-through ECRs could enhance the mass transfer of reactant by suppressing the boundary layer.

(3) From the engineering point of view, continuous mode operation is more preferred because of faster reactions, cleaner products and safer and easier scaling up than batch mode for large-scale wastewater treatment in EO system.

(4) Particle electrodes in the 3D-ECR could improve the degradation efficiency by improving the concentration of pollutants at the particle electrode interface. The 3D-ECR system deserves comprehensive development due to the particle electrodes. However, the mechanism of reactions is more complicated in a 3D-ECR. Optimizing the operating parameters and enhancing electrode materials performance of adsorption and catalysis to maintain the stable operation are significant for future development of 3D-ECRs.

At present, CFD has become an important model tool for reactor design. It could predict fluid flow, heat and mass transfer, mixing effect and electrochemical reaction mechanism in ECRs by different models. Combining simulation and practice can evaluate the flow pattern of fluid in the reactor more accurately, get more scientific and effective electrochemical performance and provide a theoretical basis for further optimization and improvement for the realization of refractory organic wastewater removal. Additionally, the removal efficiency of pollutants and energy consumption for high concentrations of refractory organic wastewater limit to the application of single EO technology. Therefore, it can also be combined with other technologies, such as membrane filtration, photochemical [90] and electrochemical technology (electroflotation [91], electrocoagulation, electrodialysis, etc.) to achieve more efficient and energy-saving removal of refractory organic wastewater. For example, the effectiveness of combined electrocoagulation (EC) with electro-oxidation (EO) for refractory organic wastewater was proven. EC as a pre-treatment could degrade various pollutants such as organic matter, suspended solids, colloidal, etc. After a certain degree of treatment, the organic matter of wastewater could be removed in an EO reactor. An amount of works showed that the EC-EO technology could not only improve the removal rate of organic matter, but also greatly reduce the energy consumption. Therefore, it may be a popular field to design and optimize combined electrochemical reactors.

**Author Contributions:** J.L.: Writing- Original draft preparation; N.R.: Literature review; C.Q.: Draw designs; S.L.: Files Preparation; Y.X.: Conceptualization; D.L.: Supervision. All authors have read and agreed to the published version of the manuscript.

**Funding:** This study was financially supported by grants from the National Natural Science Foundation of China [No. 52070008].

**Institutional Review Board Statement:** Not applicable.

**Informed Consent Statement:** Not applicable.

**Data Availability Statement:** No new data were created or analyzed in this study. Data sharing is not applicable to this article.

**Conflicts of Interest:** The authors declare no conflict of interest.

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
