# Peer review of "Recent Advances in the Reactor Design for Industrial Wastewater Treatment by Electro-Oxidation Process"

_water, doi:10.3390/w14223711_

Round 1
Reviewer 1 Report
It is necessary to review in detail the part of the spelling such as the separation of words between lines and some words that are misspelled, some of them are underlined in the attached document.
As for the content of the article, in my opinion, it is enough, as well as the discussion and presentation of several articles cited in reference to the different types of electrochemical reactors.

Author Response
Thank you for your kindly suggestions. We have changed all the spelling mistakes as you mentioned in the text marked in yellow.
Reviewer 2 Report
The current work entitled “Recent Advances in the Reactor Design for Industrial Wastewater Treatment by Electro-Oxidation Process” by Liu et al. explored different configurations of EO reactors and summarized the advantages and disadvantages of them and the advancing orientation of the reactor configurations in the future. The work done by the authors is novel and addresses important gaps in the current state of the art in reactor design in wastewater treatment. I suggest this manuscript be accepted after the following minor modifications:
1. The abstract is too general and doesn’t provide deep outcomes of the review. I suggest rewriting it and extend to 250 words.
2. The authors missed defining several abbreviations in their first use.
3. Reference formatting is not as per MDPI water.
4. Electrochemical reactions on how pollutants are breakdown by the electro-oxidation process are missing.
5. Add this report on organic load degradation: https://www.sciencedirect.com/science/article/abs/pii/S0960148121010508?via%3Dihub
6. Section 7 can be extended with more clear focus and a new heading 8. Conclusions can be added.
Reviewer 3 Report
Recent Advances in the Reactor Design for Industrial Wastewater Treatment by Electro-Oxidation Process
What is the novelty and originality of this work? Which should be clarified in the introduction
None reference from the Water journal was added, therefore it does not present relevance with this journal
Emerging trends and future prospects section should be added to the document
Real industrial application cases should be added, maybe some images of pilot plants
Therefore, I cannot recommend the submitted manuscript is published in Water in this way.
